# Pharmacological Mechanism of Ketamine in Suicidal Behavior Based on Animal Models of Aggressiveness and Impulsivity: A Narrative Review

**DOI:** 10.3390/ph16040634

**Published:** 2023-04-21

**Authors:** Thi Mai Loan Nguyen, Fabrice Jollant, Laurent Tritschler, Romain Colle, Emmanuelle Corruble, Alain M. Gardier

**Affiliations:** 1Université Paris-Saclay, Faculté de Pharmacie, Inserm CESP/UMR 1018, MOODS Team, F-91400 Orsay, France; 2Université Paris-Saclay, Faculté de Médecine, Inserm CESP/UMR 1018, MOODS Team, F-94270 Le Kremin-Bicêtre, France; 3Service Hospitalo-Universitaire de Psychiatrie, Assistance Publique-Hôpitaux de Paris (AP-HP), Hôpitaux Universitaires Paris-Saclay, Hôpital de Bicêtre, F-94275 Le Kremlin Bicêtre, France; 4Pôle de Psychiatrie, CHU Nîmes, 30900 Nîmes, France; 5Department of Psychiatry, McGill University and McGill Group for Suicide Studies, Montréal, QC H3A 0G4, Canada

**Keywords:** ketamine, suicide, antidepressant, aggressiveness, impulsivity, endophenotype

## Abstract

Around 700,000 people die from suicide each year in the world. Approximately 90% of suicides have a history of mental illness, and more than two-thirds occur during a major depressive episode. Specific therapeutic options to manage the suicidal crisis are limited and measures to prevent acting out also remain limited. Drugs shown to reduce the risk of suicide (antidepressants, lithium, or clozapine) necessitate a long delay of onset. To date, no treatment is indicated for the treatment of suicidality. Ketamine, a glutamate NMDA receptor antagonist, is a fast-acting antidepressant with significant effects on suicidal ideation in the short term, while its effects on suicidal acts still need to be demonstrated. In the present article, we reviewed the literature on preclinical studies in order to identify the potential anti-suicidal pharmacological targets of ketamine. Impulsive–aggressive traits are one of the vulnerability factors common to suicide in patients with unipolar and bipolar depression. Preclinical studies in rodent models with impulsivity, aggressiveness, and anhedonia may help to analyze, at least in part, suicide neurobiology, as well as the beneficial effects of ketamine/esketamine on reducing suicidal ideations and preventing suicidal acts. The present review focuses on disruptions in the serotonergic system (5-HT_B_ receptor, MAO-A enzyme), neuroinflammation, and/or the HPA axis in rodent models with an impulsive/aggressive phenotype, because these traits are critical risk factors for suicide in humans. Ketamine can modulate these endophenotypes of suicide in human as well as in animal models. The main pharmacological properties of ketamine are then summarized. Finally, numerous questions arose regarding the mechanisms by which ketamine may prevent an impulsive–aggressive phenotype in rodents and suicidal ideations in humans. Animal models of anxiety/depression are important tools to better understand the pathophysiology of depressed patients, and in helping develop novel and fast antidepressant drugs with anti-suicidal properties and clinical utility.

## 1. Introduction

Approximately 9% of people report having suicidal thoughts at some time in their life, and 3% actively attempt suicide [1]. Suicidal thought is 1.5 times more prevalent in women than in men, and is particularly frequent in adolescents and young adults [1]. Suicide is the leading cause of preventable mortality all over the world [2]. Every year, nearly 700,000 people in the world take their own lives, where more than two-thirds are from low- and middle-income countries. Suicide was the fourth most significant cause of death among 15–29 year olds worldwide in 2019 [3]. In addition, 10 to 20 times more people than this attempted suicide, with a significant burden at individual and collective levels. Suicidal behavior is a major public health problem worldwide.

Suicidal ideation is a prevalent symptom of many mental diseases [4]. About 90% of people who die from suicide have a history of mental disorder [5], and more than two-thirds of attempts or deaths happen during a major depressive episode (MDE) [6]. Although MDE is treatable with pharmacotherapy, up to one-third of patients do not respond to a prescribed first-line antidepressant and may remain at risk of suicide [7,8]. Furthermore, treatment-resistant depression (TRD) has been identified as a risk factor for (attempted) suicide [9]. TRD is defined as a failure of two successive antidepressant treatments of adequate dosage and duration (i.e., at least eight weeks). The estimated incidences of attempted and completed suicides in TRD patients are 4.66 and 0.47 per 100 patients per year, respectively, which are twice and ten times higher than in people suffering from non-resistant depression [10].

We recently published a review article concerning the interest of ketamine in the prevention of suicidal ideation, specifying its clinical interest [11]. It is difficult to model TRD in rodents. However, we need to develop reliable treatment response predictors to guide personalized antidepressant strategies in these patients. Thus, efforts have been made to establish a peripheral biomarker signature of response/non-response to chronic antidepressant or electroconvulsivotherapy (ECT) in animal models of anxiety/depression [12]. Thus, using a proteomic approach, the peripheral signatures of an antidepressant response (Flx-R) or non-response (Flx-NR) to chronic fluoxetine (Flx) treatment were described in the blood cells of a corticosterone (CORT) model of anxiety/depression in mice [13]. Among Flx-treated mice, one-third of them did not show a 50% decrease in emotionality score, thus defining CORT/Flx-NR. The emotionality score was based on the analysis of complementary behavioral tests of anxiety/depression [14]. Similarly, the proteomic analysis revealed a signature of 33 peripheral proteins associated with a response to ECT, which is known as a highly effective treatment that induces rapid improvement in depressive symptoms [15]. These proteins (7 down- and 26 up-regulated) are involved in regulating pathways that participate in depressive disorder etiology. Other animal models exist that represent other traits of TRD, e.g., rats exposed to maternal deprivation, which induces a depressive-like behavior. A single dose of ketamine administered during the adult phase could reverse behavioral effects and neural damage induced by this stressor in adulthood rats [16,17].

Additional risk factors for suicidal behaviors in humans are difficult to model in rodents. These include a family history of suicide (e.g., genetic factors), childhood maltreatment, recent stressful events, and access to lethal means in case of suicidal ideas, among others [18]. Some factors may contribute more to the emergence of suicidal ideas (mental disorders, mental pain, hopelessness, and ruminations), while others may facilitate the transition from suicidal ideas to an act (impulsivity/aggression, risky decision making, as well as alcohol and substance use or disorder). Moreover, some factors may be considered long-term vulnerability factors (e.g., early developmental events and certain personality traits) while others are stress/precipitating factors (e.g., recent adverse events, acute phase of a mental disorder). Overall, it is currently postulated that negative events occurring in vulnerable people may increase the risk of developing intense mental pain, which, in turn, facilitate the emergence of a suicidal state and, in some but not all people, the transition to a suicidal act.

To date, very few medications have been shown to be effective in treating acute suicidal crises or preventing suicidal acts. Since classical monoaminergic antidepressant drugs (SSRI, SNRI) display a long delay of action, rapidly acting drugs may be useful to prevent suicide and reduce the risk of suicide attempts. Can preclinical studies help analyze drugs’ beneficial effects on preventing suicidal ideation? Impulsivity may be both a trait and a state in depressed suicide attempters [19]. Impulsive–aggressive behavior is one of the vulnerability factors common to suicide, MDE, and bipolar disorder (BD) patients that can be modeled in rodents. 

Thus, this narrative review focuses first on the pathogenesis and neurobiology of suicide by describing the role of neurotransmitters (serotonin, cytokines, glutamate) and brain circuits underlying this phenotype. Then, we describe rodent models of impulsivity/aggressiveness. Finally, we summarize the pharmacological potentials of ketamine that could explain its anti-suicidal properties. As a fast-acting antidepressant, ketamine/esketamine may offer relief in the immediate term in patients with suicide ideation in TRD. The electronic database PubMed was searched for publications from 2000 to 2022, using the search terms ’suicidal behavior’ AND/OR ‘animal models of aggressiveness and impulsivity’.

## 2. The Pathogenesis and Neurobiology of Suicide

The development of suicide risks is complex; it involves multiple systems and neurological circuits with interacting contributions from genetic, biological, psychological, clinical, social, and environmental factors [20,21,22,23,24,25]. Understanding these factors might help to detect high-risk individuals and assist in treatment selection [26].

The link between disrupted serotonin expression and suicide attempts or suicide, despite the discrete differences in serotonin receptors and transporter expression between patients with depression and those who exhibit suicide ideation or suicidal behaviors, were described twenty years ago [27]. However, it is unclear to what extent altered serotonin signaling in people who die by suicide can be distinguished from changes associated with depression [26]. Indeed, these two phenotypes often occur together. Evidence shows that low serotonin is associated with personality traits linked to suicidality, such as impulsive aggression [28]. 

The dysfunction in additional neurotransmitter systems and their molecular alterations have also been implicated in depression and suicide. For example, changes in glutamate and the GABA receptor subunit gene expression in the brains of suicide victims with major depressive disorders (MDD) have yielded some initial promising results [29]. In addition, treatments for severe depression and suicide ideation that target the glutamate pathway, such as ketamine, carry much hope [4]. 

Furthermore, recent findings indicate that the pathogenesis of mood disorders and suicide involves the abnormal plasticity of neural circuits [30,31]. Indeed, it has been argued that mood disorders and suicide are caused by the brain’s inability to respond appropriately to environmental stimuli, which is due to reduced synaptic and structural plasticity. The role of consistent changes in gene expression in synaptic and structural plasticity was investigated in the postmortem brains of suicide subjects [31,32,33]. Furthermore, changes in synaptic circuitry, synaptic connectivity, and dendritic morphology have also been identified during depression and suicidality [34,35].

In addition, a meta-analysis of 18 studies comprising 1743 subjects found evidence of aberrant inflammatory cytokine (IL-1β and IL-6) levels in the blood, cerebrospinal fluid, and the postmortem brain samples of patients with suicidal behavior [36]. Thus, using the brain tissue and blood from the postmortem cohorts of MDD patients who died by suicide is of great interest in identifying at-risk patients [37]. Next, such an identification of biomarkers will require validation in prospective clinical cohorts.

Suicide neurobiology is poorly understood. The majority of the literature focuses on the brain serotonergic system, the hypothalamic–pituitary–adrenal (HPA) axis, and immune-inflammatory modulators [20]. The serotonergic system has been the most investigated and is related to suicide risk, whether directly or indirectly, through its function in mood disorders, impulsivity, and aggression [20].

## 3. Pharmacological Interventions on Suicidal Behavior

To date, very few medications have been shown to be effective in treating acute suicidal crises or preventing suicidal acts. Lithium salts (Li+) are the only mood stabilizer that has demonstrated efficacy in suicide prevention [38]. By its specific action on the serotoninergic system, treatment with Li+ significantly reduces impulsive–aggressive behavior. Long-term treatment with Li+ seems to reduce suicidal ideation, suicidal behaviors, and death by suicide in BD. The extent to which Li+ treatment prevents suicide in patients with affective disorder needs to be further investigated [39], thus reinforcing the importance of finding “anti-suicide” drug treatments. To date, no treatment has been indicated for the treatment of suicidal ideation.

The majority of pharmacological approaches regarding the treatment and prevention of suicide are focused on the treatment of psychiatric disorders [20]. Suicidal ideation frequently requires rapid intervention, yet very few treatments successfully reduce suicidal thoughts, and none are fast-acting [40]. The long delay in the clinical antidepressant efficacy observed with both pharmacotherapy (such as with SSRI-SNRI) and electroconvulsive therapy (ECT) further raises the suicidal risk [4,41]. Curbing suicidal ideation would be the ideal strategy for preventing the emergent self-harm that accounts for the majority of deaths associated with depression [4].

Recently, intravenous (IV) ketamine has been shown to reduce, within hours, the suicidal ideation in MDE and BD [11,42,43] (see below).

## 4. An Approach of Impulsivity-Aggressiveness in Rodents

A simple definition of impulsive behavior is the tendency to act prematurely without foresight. For example, such motor impulsive behavior is evident following acute morphine consumption [44]. Cognitive and psychological changes, such as a lack of impulse control, can also be observed [44]. Regulating impulsive behaviors in preclinical studies is considered in clinical conditions, and is labeled as ’impulsive-compulsive disorders’ [45]. In rodent models, the evidence generally supports a higher impulsive behavior in males when compared with females [46]. However, data from human studies are less uniform, with reports of increased impulsive action in both males and females [47].

Although animal models are essential for research, rodents do not engage in suicidal behavior that is amenable to translational studies (see [48] for a review). However, experimental research can consider the mechanisms by which drugs might be associated with potential “adverse effects” such as suicidal ideation. Many components that are implicated in the neurobiology of suicidal behaviors, as well as the neurobiology of circuits delineated as relevant to ideation in humans, may be studied in animal models. Most of them are only intended to recapitulate some of the aspects of neurobiology that are implicated in human psychiatric disorders, rather than in the disorder itself [49]. By investigating traits that show strong cross-species parallels in addition to associations with suicide in humans, animal models might elucidate the mechanisms by which drugs are associated with suicidal thinking and behavior. Traits linked with suicide in humans that can be successfully modeled in rodents include aggressiveness, impulsivity, and irritability [48]. 

Despite limitations in determining suicidal intent in animals, animal models can be used to study the neurobiological components of suicidal behaviors and its related circuits in humans. Several biomarkers (endophenotypes) have been measured in rodents that are correlated with symptoms of MDD and suicidal risk factors, at least in part because of shared underlying genetic influences. Endophenotypes such as microRNAs, neurotransmitter system abnormalities (serotonin—Table 1), HPA axis dysfunction, as well as endocrine and neuroimmune changes lead to aggressiveness, impulsivity, and decision-making deficits [50] (Table 1). The molecular pathways related to suicidal behavior can be investigated in animal models, providing insights into the neurobiology of suicidal behaviors in humans [49]. Some studies have started to identify the neural substrates and circuit-level contributions to dysregulated impulsivity [51].

MicroRNAs (miRNAs) are a type of noncoding RNA that play a role in neural plasticity and higher brain functioning [52,53] (Table 1). Several miRNAs have been found to regulate genes involved in the neurobiology of suicidal behavior [54,55]. Thus, early childhood stress is linked to the miR-16 upregulation and subsequent downregulation of the brain-derived neurotrophic factor (BDNF) gene in the hippocampus, in which depression-like behaviors were induced by maternal deprivation or chronic unpredictable stress in rats [54]. MiRNA dysregulation has also been linked to aggressive behavior and psychiatric disorders, such as bipolar disorder, depression, and schizophrenia, as well as suicide in people with no psychiatric diagnosis [56]. Smalheiser et al. (2014) showed that adaptive miRNA expression in the frontal cortex in response to shocks is blunted in rats exhibiting learned helplessness, which is a risk factor for suicidal behavior in humans [57]. The active reorganization of gene expression networks in those rats may have resulted in a persistent phenotype, thus emphasizing the importance of determining the role of miRNAs in regulating gene expression in neuronal circuits after stressful life events. 

The serotoninergic system has been involved for a long time in impulsive/aggressive behavior. In the 2000s, reducing levels of 5-HT in the central nervous system was associated with increases in impulsive behavior in both humans [58] and rodents [59] (Table 1). For example, the 5-HT system may play a role in controlling impulsive aggression through decreasing the latency to attack in male mice [60] (Table 1). Starting in the mid-1990s, genetic manipulation techniques were used to identify the specific receptor populations that modulate aggression and impulsivity in mice [61]. In the mid-2000s, conditional knockout/knockdown approaches, which selectively delete specific genes in a tissue or brain region of scientific interest, were used. For example, Htr1b null mice exhibit increased aggression and impulsivity (Table 1) [61,62]. The adolescent expression of forebrain 5-HT_1B_ heteroreceptors influences aggressive behavior; however, in a distinct set of 5-HT_1B_ receptors, it modulates impulsive behavior during adulthood [63]. In some cases, as with 5-HT_2B_R, animal models give results that converge with those obtained in humans. Further, 5Htr2b-knockout mice exhibit increased impulsivity and enhanced locomotor activity, while a particular polymorphism, i.e., a 5-HT_2B_R premature stop codon identified in the Finnish population, predisposes one to severe impulsivity [64]. By contrast, the single-nucleotide polymorphism of other members of the 5-HT system, such as the tryptophan hydroxylase genes TPH2 and the 5-HT_1A_R gene, were inconsistently linked to a higher risk of suicide [49,65]. However, the monoamine oxidase A (MAO-A) story, an enzyme that degrades serotonin and norepinephrine, is interesting. In adult transgenic mice with a MAO-A gene deletion, enhanced aggression in males was observed [66] (Table 1). Interestingly, a complete and selective deficiency of MAO-A enzymatic activity was found in five men of a Dutch family who were affected by a syndrome of borderline mental retardation and abnormal behavior including impulsive aggression [67].

Tryptophan degradation provides another hypothesis (i.e., the kynurenine pathway). L-tryptophan (L-Trp) is metabolized by the enzyme tryptophan hydroxylase to form serotonin. However, L-Trp in the body mainly breaks down into kynurenine metabolites and nicotinamide adenosine dinucleotide [68]. An imbalance in this pathway, particularly in the production of the metabolites kynurenic acid and quinolinic acid, has been associated with changes in glutamate neurotransmission and neuroinflammation, potentially contributing to mood disorders and suicidal behavior [68]. Plasma kynurenine levels were found to be elevated in suicide attempters with major depressive disorders when compared with either non-suicidal depressive patients or controls with no history of depression [69] (Table 1). However, these results have since been challenged by others [70]. In animal models of stress (foot shocks), kynurenine levels were found to be increased in the plasma and brain [71]; however, this 20-year-old data will need to be replicated. 

An additional hypothesis suggests that suicide pathogenesis is associated with neuroinflammation. Most of the influence of the brain on the immune system goes through the HPA axis (Table 1). Both stress and depression have been associated with impaired immune function. Thus, depression is now thought to be associated with the activation of some aspects of cellular immunity, resulting in the hypersecretion of proinflammatory cytokines and in the hyperactivity of the HPA axis [72] (Table 1). Evidence suggests that the underlying neurobiology of suicidal behavior involves increases in inflammatory biomarkers in both the CNS and the peripheral systems of individuals with a history of suicidal behavior. Increased cytokine levels, such as IL-4, IL-13, IL-1, IL-6, and TNF, have been reported in postmortem brain samples [24], as well as in the blood samples of patients who died by suicide or attempted suicide [73,74,75]. For example, IL-6 is elevated in the CSF of suicide attempters and is related to symptom severity [76]. In rodents, interactions between the HPA axis and the immune system were explored under stress conditions, e.g., a rat model of chronic tail restraint (Table 1). Restraints were increased in IL-1β with no changes in adrenocorticotropin (ACTH) and corticosterone (CORT) levels [77]. However, a naturalistic (predator) stressor failed to modify IL-1β, IL-1 receptor antagonist, IL-1 receptor type I, or TNF-α mRNA levels in rat prefrontal cortexes and hypothalamuses [78]. The hypothalamic expression of IL-1β and IL-2 mRNA was altered during repeated cold stress [79]. Thus, the changes in inflammatory cytokines in rodent models under stressful conditions are highly heterogeneous. It is now necessary to explore the changes in brain cytokines in rodent models of impulsivity/aggressiveness because these preclinical data are missing. One can, however, find some studies on the exacerbation of endophenotypes linked to suicidal behavior (aggression) in animal models. Interestingly, mice lacking the TNF receptor did not exhibit aggressive behavior in the resident–intruder test, suggesting that TNF also exerts a role in aggressive behavior [80]. Cytokines may have very specific effects on the brain, and further research is necessary to fully understand their impact. 

Activation of the kynurenine pathway may cause subsequent serotonin depletion and the stimulation of glutamate neurotransmission. Some changes are accompanied by decreased BDNF levels in the brain (*see below*), which is linked to impaired neuroplasticity and cognitive deficits [81] (Table 1). 

**The HPA axis/stress and glucocorticoid receptor:** It is now apparent that adaptive changes result from the chronic stress and depression that lead to a hypoactivity of the glucocorticoid receptors (GR) on immune cells and in the limbic regions of the brain (Table 1). In the late 1990s and early 2000s, questions were raised about the epigenetic modifications of the GR gene in the hippocampus by maternal behavior and licking/grooming as responses to early life stress [82] (Table 1). However, these responses have not been fully reproduced in other models in rodents [82,83]. Postmortem brain samples from suicide victims revealed altered epigenetic regulation (i.e., DNA methylation patterns) on the GR gene in the hippocampus of those with a history of childhood abuse [84,85,86]. This suggests a possible mechanism by which early life adversity may increase the risk of suicide. Lamontagne et al. (2022) also highlighted the candidate endophenotypes of suicide risks, such as sleep disturbances, changes in the 5-HT2A receptor expression associated with suicide in humans and chronic stress in animals, as well as effects of ketamine [87]. These mechanisms are complementary to those summarized here. The pathophysiology of stress is a factor involved in the onset of suicidal ideation and suicide risk. Evidence from animal models suggests that the rapid-acting mechanism of ketamine effectively rescues one from the chronic-stress-induced pathologies linked to suicide risk. Both reviews agree in emphasizing the critical need to examine the temporal dynamics of this risk factor. 

The relationship between gonadal hormones and suicidal behavior is marked by gender differences [88] (Table 1). In men, lower testosterone levels are associated with an increased risk of depression and suicide attempts, while higher testosterone is linked to increased aggression [89,90,91]. Women are more likely to attempt suicide when estrogen and progesterone levels are low [92]. Preclinical rodent studies have suggested that administering progesterone, or both estrogen and progesterone, decreased impulsivity in female rats, whereas low testosterone levels decreased impulsive behavior in male rats [46,93] (Table 1). Thus, both clinical and preclinical research agree that gonadal hormones influence suicide-related endophenotypes and the vulnerability to suicide.

**Cognitive traits—impulsivity-aggressiveness:** Indicators of aggression and impulsivity have consistently been associated with suicidal behavior and with a familial transmission of suicide behavior, i.e., “a heritable suicide phenotype” [94,95]. Suicide attempters have higher levels of lifetime impulsive aggression than those found in controls according to retrospective and prospective studies [96,97,98]. The impulsivity is likely to persist over time and may predispose people to suicidal behavior later in life [99]. Impulsive aggression can also be evaluated in rodent behavioral tasks such as decreased inhibitory control, inability to delay reward, and impaired decision making. 

After the above, the role of ketamine in these developments starts. 

## 5. Main Pharmacological Properties of Ketamine

***(R,S)-, (S)-* and *®-*ketamine:** *(R,S)-*ketamine was first described in 1965 and used as a general anesthetic and veterinary analgesic drug since 1970 [100,101,102,103]. *(R,S)-*ketamine has multiple properties and indications, e.g., for both acute and chronic pain treatment [104]. 

A 2019 meta-analysis reported that IV *(R,S)-*ketamine (0.5 mg/kg infused over 40 min) reduced suicidal ideation, but no evidence of efficacy was found for other routes of administration [105]. In 2019, an intranasal (IN) spray of esketamine was approved, along with oral antidepressant drugs, as a fast-acting antidepressant by the drug agencies US FDA and European EMA [106,107]. In this indication, esketamine, in combination with an SSRI or an SNRI, constitutes an alternative for adult patients under 65 years of age for the treatment of characterized resistant major depressive episodes—if they are severe and/or a contraindication, or resistant to ECT—who have not responded to at least two different classes of antidepressant drugs. It can be prescribed in the USA for TRD in adults, as well as to treat depressive symptoms in adults with MDE and acute and severe suicidal ideation. However, IN esketamine has moderate effects on depression *versus* a placebo, and it has not shown convincing effects on suicidal ideation to date. Indeed, a recent review pointed out that the results from clinical trials did not demonstrate the antisuicidal effects of esketamine in patients with baseline suicidality [108]. By contrast, ketamine rapidly decreased the severity of suicide ideation and depressive symptoms, but these effects were short-lasting. 

Since its discovery in the sixties, many pharmacological properties of ketamine (binding to a glutamate NMDAR, signaling pathways, etc.) have been described. Here is a short summary of them:

*(R,S)-*ketamine is a racemic mixture containing equal parts of (*R*)-ketamine (or arketamine) and (*S*)-ketamine (or esketamine). *(R,S)-*ketamine is a non-competitive antagonist of the *N*-methyl-D-aspartate receptor (NMDA-R). (*R*)-ketamine (or arketamine) [109] has displayed longer-lasting antidepressant-like effects than (*S*)-ketamine in preclinical models of anxiety/depression mice [110,111]; this was despite its four times lower affinity to NMDA-R when compared to (*S*)-ketamine [111]. (*S*)-ketamine is considered to have twice the therapeutic index of racemic ketamine, which means that theoretically adverse effects should be reduced when only half of the usual racemic dose is administered [112]. However, a direct comparison between esketamine (S-ketamine), arketamine (R-ketamine) enantiomers, and racemic ketamine, specifically regarding their antidepressant efficacy in animal models [113] and in clinical trials are still necessary [114,115,116]. 

The pharmacological properties of *(R,S)-*ketamine describe a powerful NMDA-R antagonist in vitro, EC_50_ = 760 nM *in vivo*, and ED_50_ = 4.4 mg/kg in rodents’ cortex or hippocampus [117,118]. Glutamate is the major excitatory neurotransmitter in the central nervous system. Under physiological conditions, and in the presence of Mg^2+^, the NMDA channel is closed, which theoretically should inhibit the ability of *(R,S)-*ketamine to bind to its phencyclidine-like site. However, an electrophysiological *in vitro* study performed in cultured hippocampal neurons has shown that *(R,S)-*ketamine can still block NMDA-R and reduce post-synaptic currents under physiological conditions (i.e., in the presence of Mg^2+^) [119]. These data suggest that *(R,S)-*ketamine readily exceeds the physiologic capacity of the NMDA-R’s Mg^2+^-dependent voltage gating to impede ion flow through the receptor channel. 

**NMDA-R subunits:** *(R,S)-*ketamine blocks NMDAR on GABA-Gad1 interneurons as well as the subtypes of interneurons expressing somatostatin (Sst) or parvalbumin (Pvalb) in order to induce glutamate release and the indirect activation of excitatory synapses [120,121]. Behavioral studies using cell-type-specific knockdown in the medial prefrontal cortex (mPFC) demonstrated that NMDAR-GluN2B knockdown on Gad1-expressing neurons blocks the antidepressant effects of ketamine in mice [121]. These results provide evidence for initial cellular triggers, neuronal cell types, and the specific NMDA-GluN2B receptor subtype involved in the antidepressant actions of *(R,S)-*ketamine.

Another NMDA-R antagonist, such as memantine, does not inhibit the phosphorylation of eukaryotic elongation factor-2 (eEF2) kinase or increase the BDNF protein expression, which are the essential determinants of ketamine-mediated antidepressant effects [119].

**(*R,S*)-ketamine and AMPA receptor:** The downstream activation of the AMPA receptor (AMPA-R) seems to play an important role in the prefrontal cortex and hippocampal neurons, thus leading to fast ketamine-induced antidepressant effects. NBQX, an AMPA-R antagonist, blocks the neurochemical and behavioral effects of a single, subanesthetic dose of ketamine [122,123]. However, (*R*)-ketamine and (*S*)-ketamine remodel neurotransmission and connectivity through different mechanisms. (*R*)-ketamine strongly activates the prefrontal serotonergic system through an AMPA-receptor-independent mechanism, while (*S*)-ketamine-induced serotonin and dopamine release are activated through the AMPA-receptor-dependent mechanism [124]. These findings provide a neurochemical basis for the underlying pharmacological differences between ketamine enantiomers and their metabolites. 

**(2*R,6R*)-hydroxynorketamine, the primary brain metabolite of *(R,S)-*ketamine:** (*2R,6R*)-HNK would have antidepressant-like effects via a mechanism that does not involve a direct inhibition of NMDA-R but for its potential binding to AMPA-R and AMPA-R agonist properties [125]. *(R,S)-*ketamine rapidly and stereoselectively metabolizes into several metabolites, e.g., (*S*)-ketamine is metabolized to (*S*)-norketamine and (*2S,6S*; *2S,6S*)-HNK (in human: [112]; in rodents: [126,127]). Preclinical studies have indicated that HNKs display antidepressant-like activity in rodent models of anxiety/depression [125]. However, the pharmacological properties of (2*R,*6*R*)-HNK and its contribution to the actions of (*R,S*)-ketamine are still up for debate [128,129,130,131,132,133]. We are waiting for the results of clinical trials regarding (2*R,*6*R*)-HNK and its potential therapeutic applications in TRD [131]. 

**The opioid receptor activation hypothesis:** In the 1980s, the role of mu, kappa, and sigma opiate/phencyclidine (PCP) receptors was investigated [134,135]. Recently, Williams et al. (2018) showed that ketamine’s acute antidepressant effect requires opioid receptor activation in adults with TRD [136]. Furthermore, this action may play a role in the anti-suicidality effects of ketamine [137]. This hypothesis attracted attention, but these two studies involved only 12–14 patients who had received one pre-treatment dose of naltrexone. Although the dissociative effects of ketamine do not seem to be mediated by the opioid receptor signaling [138], further studies are requested before drawing a solid conclusion. Importantly, George (2018) emphasizes that “*TRD is a chronic and recurring illness, and little is known whether ketamine may become a chronic treatment*” [139]. 

**The BDNF and mTOR/TrkB receptors:** Beyond NMDA-R antagonism, it has been shown that downstream mechanisms regulating synaptic plasticity involve a cascade of postsynaptic intracellular events that are activated by *(R,S)-*ketamine, which increases the phosphorylation of the mechanistic target of rapamycin (mTOR) [140,141,142] and the expression of synaptic proteins (eEF2 kinase, BDNF, its high-affinity TrkB receptor, synapsin 1, PSD95, etc.). These findings suggest that the activation of the mTOR signaling pathways is necessary for the ketamine-induced synaptogenesis in mPFC pyramidal neurons, usually 24 h after a single *(R,S)-*ketamine dose in rodents [122]. Several studies performed in rodents used rapamycin (or sirolimus, intracerebroventricular, i.c.v., 10 mg/mL, 2 µL), which is an mTORC1 inhibitor of rapamycin complex 1, to block the antidepressant effects of ketamine [122,142]. However, a single dose of rapamycin failed to achieve this in depressed patients [143]. These discrepancies suggest that an i.c.v. injection of rapamycin is requested to affect a network in the prefrontal /hippocampal circuit. In summary, *(R,S)-*ketamine may exert acute changes in synaptic plasticity, leading to a sustained strengthening of excitatory synapses, thus leading to its fast antidepressant action [144,145]. 

**mPFC vs. hippocampus:** In the mPFC and hippocampus, ketamine blocks NMDA-R, which ultimately induces the activation of BDNF release and therefore the activation of its high-affinity receptor TrkB. However, between these two brain regions, the mechanisms seem to differ. In the hippocampus, an NMDA-R blockade by ketamine induces a deactivation of eEF2 kinase (also called CaMKII, which suppresses BDNF release), which is necessary for the de-repression of BDNF, post-synaptic AMPA-R activation, and ketamine antidepressant-like activity [146]. By contrast, the signaling cascade induced by ketamine in the mPFC goes through the activation of mTOR, then to BDNF release [122,147]. A link between ketamine mTOR in the hippocampus, or eEF2 in the mPFC, is currently missing. Lisa Monteggia’s group recently suggested that the deactivation of eEF2 kinase is requested for the initiation of the ketamine antidepressant effect [146]. It is possible that despite different signaling pathways being involved in the effects of ketamine in the cortex and hippocampus, they could converge to synaptogenesis (a kind of synergistic effect in the cortex–hippocampus circuitry).

**Excitatory/inhibitory balance:** The balance of excitation/inhibition synaptic transmission (E/I balance) is now under investigation in mPFC and in the hippocampus for two main reasons: 

i—neurophysiology-pathology: this is its role in the regulation of the cortex–hippocampus neurocircuit function in rodents [148] and its dysregulation in various brain pathologies (schizophrenia: [149]; cognition: [150]; anxiety: [151]). For example, the E/I balance is known to regulate the function of cortex–hippocampus neurocircuits in rodents [152]. The E/I imbalance can result in synaptic deficits, which play an essential role in depression/anxiety [153]. Thus, changes in E/I balance may be a biomarker of the pathophysiology developed in rodent models of anxiety/depression. 

ii—pharmacology: E/I balance can be modified by a pharmacological treatment, thus being a biomarker of drug efficacy. We recently found an imbalance of excitatory to inhibitory input toward increased excitatory activity in the mPFC of ketamine-treated mice [154]. Our data agree with Huang and Liston (2017) [155], showing a rapid and transient increase in global glutamate activity (i.e., glutamate release by mPFC glutamatergic axon terminals) in the cortex following ketamine administration.

Over the past decade, accumulating evidence has revealed that dysfunctions in the E/I balance in the mPFC are implicated in the pathology of depression [156,157]. Evidence from the literature has demonstrated that parvalbumin-positive interneurons (PV), the most abundant subtype of the total population of cortical GABAergic interneurons, are the major regulators of E/I balance in the mPFC. Depending on the route of administration (intravenous, intraperitoneal, intranasal), *(R,S*)-ketamine may influence brain E/I balance through a potentiation of glutamatergic synaptic excitation, thus inducing a sustained antidepressant-like brain state [151,158,159]. The divergent pharmacological modulatory effects of (R)*-*ketamine, *(S)-*ketamine, and *(R,S)-*HNK metabolites on the E/I imbalance in various brain regions are still a matter of debate [138,160]. 

**The disinhibition hypothesis of pyramidal cells in the prefrontal cortex:** There are several hypotheses for ketamine’s mechanism of action. The main hypothesis is the disinhibition of pyramidal cells via a decreased output of fast-spiking GABAergic interneurons in the mPFC and hippocampus [120,161,162,163]. It has been proposed as a key mechanism that triggers *(R,S)-*ketamine’s antidepressant response. A low dose of *(R,S)-*ketamine blocks the NMDA-Rs located on GABA interneurons in the mPFC, resulting in decreased inhibitory inputs and increased small excitatory post-synaptic currents (sEPSCs) on layer V neurons [120] or in increased glutamate release in the mPFC [164]. However, different in vitro/in vivo protocols have been used to test this hypothesis. When using in vitro electrophysiology in the dorsal hippocampus slices in naive rats, ketamine reduced the inhibitory input onto pyramidal cells and increased the synaptically driven pyramidal cell excitability at the single-cell and population levels [163]. By contrast, when using in vivo microdialysis in a mouse model of anxiety/depression, ketamine increased GABA release in the mPFC at 24 h post-administration [165]. Chronic stress yielding a depressed phenotype may explain these discrepancies [166]. The route of ketamine administration could be another important factor. We recently found that the intranasal administration of (*R,S*)-ketamine induced a dose-dependent increase in cortical glutamate release, but not in a GABA release as opposed to a systemic ketamine dose, suggesting that ketamine metabolites produced in the liver following an IP injection may induce a GABAergic action [154]. Furthermore, major depression is associated with low plasma and cerebrospinal fluid GABA concentrations in depressed patients [167]. Thus, this hypothesis needs to be further investigated by comparing the E/I balance in the mPFC versus the hippocampus. The role of the connectivity between the ventral hippocampus and mPFC was recently highlighted using the optogenetic activation of this circuit [168]. 

## 6. Pharmacological Potentials of Ketamine Action in Suicide Behavior

Early studies in 2014 used pharmacologic approaches that have been considered potential treatments to reduce suicide risk, namely by reducing suicide deaths, attempts, and ideation [40]. The protective effects of lithium salts, clozapine, antidepressants, antipsychotics, and ECT were evaluated to decrease the risk of engaging in suicidal behavior [49]. However, despite inconsistent findings regarding drug efficacy in the reduction in suicidality in both unipolar and bipolar depression [49], the discovery of new targets for the treatment of suicide ideations is necessary. We need to better understand the underlying pathophysiology of suicidal thoughts and the brain regions/circuits/connectivity involved in responses to *(R,S)-*ketamine. 

Although ECT is a gold-standard therapy for TRD, ketamine has arisen as an (experimental) alternative over the last two decades with the benefit of rapid onset action and relief from depressive, anxious, and anhedonic symptoms, including suicidal ideations [169]. The short-term efficacy of ketamine in TRD has been confirmed in randomized, double-blind, placebo-controlled trials [170]. Patients with MDD having active suicidal ideation with intent require immediate treatment. A single IV subanesthetic dose of ketamine is assumed to have rapid effects on suicidality in TRD, and acute improvements in suicidality can be sustained with repeated ketamine infusions [171]. 

Recently, two clinical trials demonstrated the rapid and robust efficacy of an esketamine nasal spray in reducing depressive symptoms in severely ill patients with major depressive disorder who were having active suicidal ideation with intent [172,173]. A meta-analysis of five randomized, double-blind clinical trials with 744 MDD patients with TRD comparing adjunctive IN esketamine to an adjunctive placebo confirmed the rapidity and efficacy of esketamine (with respect to MADRS score change, response, and remission) in treating an imminent risk of suicide [174]. Strikingly, its anti-suicidal and antidepressant benefits in patients with suicidal ideations may be independent of one another. Esketamine’s antidepressant activity may last longer than its anti-suicide effect [175]. Another meta-analysis corroborates the efficacy of both IN and IV ketamine administration for the treatment of MDD, but suggests no difference between either route of delivery [115]. However, this assertion was challenged [116]: In the absence of head-to-head studies directly comparing the efficacy of IN esketamine to that of IV ketamine, valid conclusions regarding comparative efficacy should be taken with caution. Existing data from clinical trials were obtained by using markedly differing study designs and patient populations.

The fast-acting antidepressant activity of *(R,S)-*ketamine as a glutamatergic NMDA-R antagonist makes it a valuable research tool for understanding the neurobiology of antidepressant and anti-suicidal responses. “Old data” suggest that *(R,S)-*ketamine may have anti-aggressive properties [176], but this requires further investigation. Disruptions in the glutamate–glutamine cycle were found in the underlying pathology of multiple suicide-related psychiatric conditions such as major depressive disorder [177]. Recently, studies performed in experimental animal models demonstrated that glutamate plays a key role in aggressiveness and impulsivity [177,178]. Thus, efforts to describe neural correlates of suicidal ideation must continue.

Little is known about the mechanism of action and the biomarkers of the response to *(R,S)-*ketamine in the treatment of suicidal thoughts (see [179] for a review). Knowing that the dose of *(R,S)-*ketamine administered in patients with TRD is similar to that used in patients with suicidal ideation (0.5 mg/kg intravenously perfused over 40 min), it can be assumed that the treatment of suicidality with ketamine potentially goes through similar signaling pathways, i.e., the disinhibition hypothesis, synaptogenesis, and the BDNF/mTOR/eEF2 pathways behind NMDA-R/AMPA-R. 

## 7. Conclusions

In summary, whether *(R,S)-*ketamine-esketamine effects on impulsivity and aggressiveness are relevant to suicidal behavior remains to be confirmed in preclinical studies. Identification of the mechanisms of pathophysiology, suicide risk factors of neuropsychological vulnerabilities, cognitive deficits, the endophenotypes of sub-groups of patients presenting with suicide ideations/suicidality, as well as the role of critical interactions between genetic and epigenetic factors associated with suicidal behavior may help to solve this issue. Studies of deficits in the balance of motivational and inhibitory processes may be key factors to solve these issues [180]. In addition, neural and behavioral motivational processes highlight the importance of reward circuitry in ketamine’s mechanism of action in motivational symptoms [181]. Further studies should be conducted to identify other neural substrates and circuit-level contributions to dysregulated impulsivity. Thus, anhedonia was associated with specific neural circuits in response to *(R,S)-*ketamine, and suicidal ideation may be related to symptoms of anhedonia that are independent of depressive symptoms [182]. 

The present review should be read in light of the following limitations. Some endophenotypes have been selected (i.e., the 5-HT system and cytokines), but the list is not exhaustive. Certainly, ketamine is the first new treatment for depression in 60 years, with an original mechanism of action, but its effects on suicidal ideation need to be confirmed. To date, there are no studies on the risk/prevention of suicidal acts and mortality, and the long-term adverse effects, particularly the influence of events of abuse, are unknown [41]. Due to the dissociative/psychotomimetic effects of ketamine, the need for its IV infusion requires an acute hospitalization of patients with MDD. 

Regarding animal models, efforts must focus on the biomarker identification of (*R,S)-*ketamine response in rodents, including electrophysiological and imaging studies; metabolic, immune, and neurotrophic biomarkers; and the role of neuroinflammation and microglial cells. None of these biomarkers are ready for clinical use in the subgroups of depressed patients with suicidal ideations, nor in those having a co-morbidity, such as those with anxiety, neuroinflammation, metabolic symptoms [183].

However, a key question still remains regarding the potential abuse liability of *(R)-*ketamine versus *(S)-*ketamine (in rodents: [138]). Is it necessary to limit the ketamine affinity at opioid receptors and the dopamine tone in various brain regions (mPFC, nucleus accumbens, basal ganglia) to reduce its adverse effects? (i.e., a favorable risk/benefit ratio). Before being engaged in the second generation of serotonergic and/or glutamatergic antidepressant drugs, e.g., psilocybin (a 5-HT_2A_ receptor agonist), additional studies must be undertaken to examine the dopaminergic activity linked to addiction, abuse liability, and drug dependency [184]. Understanding the pharmacology of *(R,S)-*ketamine and its enantiomers could allow the development of analogous, but safer drugs [185]. Thus, there are concerns about safety and long-term outcomes of glutamatergic antidepressant drugs because the benefit/risk ratio is a relevant question in clinical practice [172]. In the future, there should be more specific molecules, with direct or indirect glutamatergic mechanisms of action for efficacy, but with fewer adverse effects.

**Table 1 pharmaceuticals-16-00634-t001:** Endophenotypes of suicidal behavior as studied in animal models and humans.

Targets	Rodents	Humans
MicroRNAs	MiRNA expression in the frontal cortex following shocks is blunted in rats exhibiting learned helplessness [57].	MiRNA dysregulation is linked to aggressive behavior and psychiatric disorders, as well as to suicide in people with no psychiatric diagnosis [56].
L-Trp and serotonin system	Extracellular 5-HT and 5-HIAA levels in the mPFC are negatively correlated with levels of aggression in rats [59].	Low CSF 5-HIAA levels and the dysregulation of the 5-HT system are associated with aggression and impulsivity in humans [58].
L-Trp and kynurenine pathway	Plasma and brain kynurenine levels are increased by foot shocks in rats [71].	Plasma kynurenine levels are higher in the CSF of suicide attempters than in healthy people [69,81].
MAO-A	MAO-A deficiency is associated with aggressive behavior [66].	MAO-A deficiency is associated with the disturbed regulation of impulsive aggression [67].
5-HT_1B_ receptor	Htr1b null mice exhibit increased aggression and impulsivity [61,62].5-HT_1B_R knockout mice show increased aggression and impulsivity, and 5-HT_1B_R polymorphisms are associated with aggression [63].	Adult aggression is determined by forebrain 5-HT_1B_ heteroreceptors.Impulsive behavior is modulated by a different population of 5-HT_1B_ heteroreceptors [186].
BDNF	Heterozygous BDNF^+/−^ mice, conditional BDNF knockout mice, and mutant mice with a specific disruption of *Bdnf* promoter I exhibited enhanced inter-male aggression [187].	BDNF protein levels in the prefrontal cortex, hippocampus, and amygdala are lower in the postmortem brain of individuals who died by suicide [188]. BDNF Val/Met is related to the suicidal behavior of bipolar patients [189].
HPA axis	Increased maternal licking/grooming is linked to an attenuation of the response to stressful stimuli in rats [82].	DNA methylation patterns on the GR gene are altered in the hippocampus of suicide victims with a history of childhood abuse [84,85].
Cytokines	IL-1, IL-2, and TNF increase aggression in rodents [80,190,191,192].	IL-4, IL-13, IL-1, IL-6, and TNF increase in the postmortem brain samples and blood samples of patients who died by suicide or attempted suicide [73,74,75].
Glutamate	Extracellular glutamate clearance decreases in nandrolone decanoate-induced aggressive behavior in mice [193].	The AMPA glutamate receptor subtype increases in the caudate nucleus of schizophrenics and nonpsychotic suicides [194].
Gonadal hormones	Administering progesterone or both estrogen and progesterone decreases impulsivity in female rats. Low testosterone levels decrease impulsive behavior in male rats [46,93].	Lower testosterone levels increase the risk of depression and suicide attempts. Higher testosterone levels increase aggression in men [89,90,91]. Lower estrogen and progesterone levels are linked to attempted suicide in women [92].
Microbial pathogens	Latent T. gondii infection reduces and reverses the innate fear of cat odor and other stimuli that precede predation [195].	Suicidal behavior, aggression, and impulsivity are linked to chronic T. gondii infection in both healthy people and psychiatric patients [196,197].

## Data Availability

Not applicable.

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
