# Peer review of "Pharmacological Mechanism of Ketamine in Suicidal Behavior Based on Animal Models of Aggressiveness and Impulsivity: A Narrative Review"

_pharmaceuticals, 2023, doi:10.3390/ph16040634_

Round 1

Reviewer 1 Report

The subject of this review is very interesting and worth investigating although I have a few remarks

The review is a bit chaotic going from studies on impulsivity and aggression to anhedonia and suicidality and the links are not clear enough. The form like "etc..." in such a review should not be used, the authors should be more precise in the content they want to present to the readers.

Esketamine is registered not only in the US but also in Europe by EMA; this should be mentioned.  

The authors should discuss the difference in antisuicidal effects of IN esketamine and IV ketamine, look below:

Siegel AN, Di Vincenzo JD, Brietzke E, Gill H, Rodrigues NB, Lui LMW, Teopiz KM, Ng J, Ho R, McIntyre RS, Rosenblat JD. Antisuicidal and antidepressant effects of ketamine and esketamine in patients with baseline suicidality: A systematic review. J Psychiatr Res. 2021 May;137:426-436. doi: 10.1016/j.jpsychires.2021.03.009. Epub 2021 Mar 16. PMID: 33774537.

The review lacks the most recent studies on the antisuicidal effect of ketamine.

Lamontagne SJ, Ballard ED, Zarate CA Jr. Effects of stress on endophenotypes of suicide across species: A role for ketamine in risk mitigation. Neurobiol Stress. 2022 Apr 20;18:100450. doi: 10.1016/j.ynstr.2022.100450. PMID: 35685678; PMCID: PMC9170747.

Liu H, Wang C, Lan X, Li W, Zhang F, Fu L, Ye Y, Ning Y, Zhou Y. Functional connectivity of the amygdala and the antidepressant and antisuicidal effects of repeated ketamine infusions in major depressive disorder. Front Neurosci. 2023 Feb 2;17:1123797. doi: 10.3389/fnins.2023.1123797. PMID: 36816116; PMCID: PMC9932998.

A graphic presentation of the results would be useful. 

The authors do not mention that ketamine is investigated in TRD and animal models are not really sufficient for this kind of depression, this should be discussed, look below:

Réus GZ, de Moura AB, Borba LA, Abelaira HM, Quevedo J. Strategies for Treatment-Resistant Depression: Lessons Learned from Animal Models. Mol Neuropsychiatry. 2019 Oct;5(4):178-189. doi: 10.1159/000500324. Epub 2019 May 21. PMID: 31768371; PMCID: PMC6873047.

It is not clear what is the reason for concentrating on ketamine's enantiomers so much. I get an impression of a lot of information is not really connected, so the paper needs to work on the cohesion of content.

The conclusions refer only to ketamine, which is just a part of this paper. There is no reflection of the first part in this section.

It is hard to follow what is the main idea the authors want to present.

Concentrating on the main plot and sticking to it would make the article much clearer.

I encourage authors to read more papers in order to get a more clear view since the subject is definitely interesting and worth writing about.

Author Response

Reviewer#1 :

We would like to thank the reviewer for his/her few remarks. As requested, we removed “etc…” all over the manuscript.

1- Esketamine is registered not only in the US but also in Europe by EMA; this should be mentioned.

Yes, esketamine is also registered in Europe by EMA as it was already indicated on page 14 (now page 16).

2- The authors should discuss the difference in antisuicidal effects of IN esketamine and IV ketamine, look below:

We discussed differences between anti-suicidal effects of IN esketamine and IV ketamine and added the review of Siegel et al., 2021 in the references’ list. On page 16, lines 9-12, we added: Indeed, a recent review pointed out that results from clinical trials did not demonstrate antisuicidal effects of esketamine in patients with baseline suicidality (Siegel et al., 2021). By contrast, ketamine rapidly decreased the severity of suicide ideation and depressive symptoms, but these effects were short-lasting.”

3- The review lacks the most recent studies on the antisuicidal effect of ketamine.

Sorry, among 197 references we quoted, we missed some recent reviews on ketamine in preclinical (animal models) and clinical studies and its stress-related effects on suicide risk. However, we previously dedicated a paragraph on HPA axis/stress on page 14 to underline the need to target the HPA axis/stress system in suicide prevention.

On page 14, we now discuss this review as follows: Lamontagne et al., (2022) also highlighted candidate endophenotypes of suicide risks such as sleep disturbances, changes in 5-HT2A receptor expression associated in suicide in humans and chronic stress in animals as well as effects of ketamine. These mechanisms are complementary to those summarized here. Pathophysiology of stress is a factor involved in the onset of suicidal ideation and suicide risk. Evidence from animal models suggests that the rapid-acting mechanism of ketamine effectively rescues chronic stress-induced pathologies linked to suicide risk. Both reviews agree in emphasizing the critical need to examine the temporal dynamics of this risk factor.”

However, in their review, Lamontagne et al., (2022) did not mention important studies on impulsive/aggressive phenotype in animal models of serotonergic system alterations such as 5-HT1B or MAOA KO mice as we do here: thus, our two reviews are complementary.

  1. A graphic presentation of the results would be useful. 

Our Table 1 summarized our results.

  1. The authors do not mention that ketamine is investigated in TRD and animal models are not really sufficient for this kind of depression, this should be discussed.

We already mentioned that ketamine is investigated in TRD on page 16, line 6. TRD characterized resistant major depressive episodes which have not responded to at least two different classes of antidepressant drugs (on page 5).

Yes, it is difficult to model TRD in rodents, but not impossible. However, it is necessary to develop reliable treatment response predictors to guide personalized antidepressant strategies in these TRD patients. Thus, efforts have been made to establish a peripheral biomarkers signature of response/non-response to chronic antidepressant or electroconvulsivotherapy (ECT) in animal models of anxiety/depression (Mendez-David et al., 2017). Thus, using a proteomic approach, peripheral signatures of antidepressant response (Flx-R) or non-response (Flx-NR) to a chronic fluoxetine (Flx) treatment were described in blood cells in the corticosterone (CORT) model of anxiety/depression in mice (David et al., 2009). Among Flx-treated mice, one-third of them did not show a 50% decrease in emotionality score, defining CORT/Flx-NR. The emotionality score is based on the analysis of complementary behavioral tests of anxiety/depression (Guilloux et al., 2011). Similarly, the proteomic analysis revealed a signature of 33 peripheral proteins associated with response to ECT, which is known as a highly effective treatment inducing rapid improvement in depressive symptoms (Lebeau et al., 2022). These proteins (7 down and 26 upregulated) are involved in regulating pathways which participate in the depressive disorder etiology.

We agree with Réus et al., (2019) saying that animal models of anxiety/depression (not depression) are important tools to better understand the pathophysiology of TRD patients. These models represent some traits of TRD (as impulsivity/aggressiveness), and help in understanding the mechanism by which ketamine, for example, exert its antidepressant effects via the regulation of neurotransmitter activity.

On page 6, we wrote: Other animal models exist representing traits of TRD, e.g. rats exposed to maternal deprivation, which induces a depressive-like behavior. A single dose of ketamine administered during the adult phase was able to reverse behavioral effects and neural damage induced by this stressor in adulthood rats (Réus et al., 2015)”.

We discussed these points on pages 5-6 of the manuscript.

  1. It is not clear what is the reason for concentrating on ketamine's enantiomers so much.

We strongly believe that the future of the ketamine’s story will be its enantiomers (despite the cost of the treatment and Siegel et al., 2021). Pharmaceutical companies known the rules of the FDA/EMA agencies: when they develop a racemic mixture, they have to separate the enantiomers and show which one has better efficacy and safety.

  1. The conclusions refer only to ketamine, which is just a part of this paper. There is no reflection of the first part in this section.

We changed the Conclusion accordingly.

  1. It is hard to follow what is the main idea the authors want to present.

Sorry if our idea wasn’t explained clearly. We recently published a first review regarding the clinical interest of ketamine in the prevention of suicidal ideation (Jollant et al., 2022). The present study is the second part of our work focusing on preclinical studies performed in animal models of impulsivity/aggressiveness.

We modified the Abstract accordingly: “The present review focuses on disruptions in the serotonergic system (5-HTB receptor, MAO-A enzyme), neuroinflammation and/or HPA axis in rodent models with an impulsive/aggressive phenotype because these traits are critical risk factors for suicide in humans. Ketamine can modulate these endophenotypes of suicide in humans as well as in animal models.”

  1. I encourage authors to read more papers in order to get a more clear view since the subject is definitely interesting and worth writing about.

We would like to thank the reviewer for his/her advice.

Reviewer 2 Report

This manuscript describes pharmacological mechanism of ketamine in suicidal behavior based on animal models of aggressiveness and impulsivity in an  narrative manner. The work is important and well-organized and comprehensively described, although a bit of typos can be found. 

The clinical investigations of ketamine on behavioral effects remain sparse yet. Therefore , this review will provide an update for interested readers to understand current evidence on biological and pharmacological mechanism of ketamine in suicidal behavior.

Author Response

  1. This manuscript describes pharmacological mechanism of ketamine in suicidal behavior based on animal models of aggressiveness and impulsivity in a narrative manner. The work is important and well-organized and comprehensively described, although a bit of typos can be found. 

We corrected the typos accordingly.

  1. The clinical investigations of ketamine on behavioral effects remain sparse yet. Therefore, this review will provide an update for interested readers to understand current evidence on biological and pharmacological mechanism of ketamine in suicidal behavior.

Thank you very much for your comments.

Round 2

Reviewer 1 Report

1. There is no cntinuation between subseqient paragraphs:

"Other animal models exist representing other traits of TRD, e.g. rats exposed to maternal deprivation, which induces a depressive-like behavior. A single dose of ketamine administered during the adult phase could reverse behavioral effects and neural damage induced by this stressor in adulthood rats [15,16]

Additional risk factors for suicidal behaviors are a family history of suicide (including genetic factors), childhood maltreatment, impulsive and aggressive personality traits, recent stressful events, and access to a lethal means in case of suicidal ideas, among others [17]. "

2. What do you mean by "this is a short review" do you mean it covers just specific part of the subject? Maybe better would be to call it a narraticie review.

3. I think it would also be useful to include the limitation section

4. The description of methodology should be included describing how the search was performed.

Author Response

Reviewer#1 :

Thank you for your last remarks.  

  1. There is no continuation between subsequent paragraphs:

"Other animal models exist representing other traits of TRD, e.g. rats exposed to maternal deprivation, which induces a depressive-like behavior. A single dose of ketamine administered during the adult phase could reverse behavioral effects and neural damage induced by this stressor in adulthood rats [15,16]. 

Additional risk factors for suicidal behaviors are a family history of suicide (including genetic factors), childhood maltreatment, impulsive and aggressive personality traits, recent stressful events, and access to a lethal means in case of suicidal ideas, among others [17]."

On page 6, we added the following sentence: “Additional risk factors for suicidal behaviors in humans are difficult to model in rodents. It includes a family history of…..”.

  1. What do you mean by "this is a short review" do you mean it covers just specific part of the subject? Maybe better would be to call it a narrative review.

On page 7, we modified accordingly: “a narrative review”

  1. I think it would also be useful to include the limitation section.

On page 24, we added the following paragraph: “The present review sould be read in light of the following limitations. Some endophenotypes have been selected (the 5-HT system, cytokines), but the list is not non-exhaustive. Certainly, ketamine is the 1st new treatment for depression in 60 years, with an original mechanism of action, but its effects on suicidal ideation need to be confirmed. To date, there are no studies on the risk/prevention of suicidal acts and mortality and the long-term adverse effects, in particular the event of abuse, are unknown [41]. Due to the dissociative/psychotomimetic effects of ketamine, its i.v. infusion requests an acute hospitalization of patients with MDD. Regarding animal models, …”

4.The description of methodology should be included describing how the search was performed.

On page 7, we added the following sentence: “The electronic database PubMed was searched for publications, using the search terms “’suicidal behavior’ AND/OR ‘animal models of aggressiveness and impulsivity’ from 2000 to 2022.

Thank you again for your comments.
